# Variation of Complement Protein Levels in Maternal Plasma and Umbilical Cord Blood during Normal Pregnancy: An Observational Study

**DOI:** 10.3390/jcm11133611

**Published:** 2022-06-22

**Authors:** Muna Saleh, Michele Compagno, Sofia Pihl, Helena Strevens, Barbro Persson, Jonas Wetterö, Bo Nilsson, Christopher Sjöwall

**Affiliations:** 1Department of Biomedical and Clinical Sciences, Division of Inflammation and Infection/Rheumatology, Linköping University, SE-581 85 Linköping, Sweden; jonas.wettero@liu.se (J.W.); christopher.sjowall@liu.se (C.S.); 2Rheumatology, Department of Clinical Sciences Malmö, Faculty of Medicine, Lund University, SE-222 42 Lund, Sweden; michele.compagno@med.lu.se; 3Department of Obstetrics and Gynaecology, Linköping University Hospital, SE-581 85 Linköping, Sweden; sofia.pihl@regionostergotland.se; 4Department of Biomedical and Clinical Sciences, Division of Children’s and Women’s Health, Linköping University, SE-581 85 Linköping, Sweden; 5Department of Clinical Sciences Lund, Department of Obstetrics and Gynaecology, Lund University, SE-222 42 Lund, Sweden; helena.strevens@med.lu.se; 6Rudbeck Laboratory C5:3, Department of Immunology, Genetics and Pathology, Uppsala University, SE-751 85 Uppsala, Sweden; barbro.persson@igp.uu.se (B.P.); bo.nilsson@igp.uu.se (B.N.)

**Keywords:** pregnancy, complement system, umbilical cord blood

## Abstract

The complement system constitutes a crucial part of the innate immunity, mediating opsonization, lysis, inflammation, and elimination of potential pathogens. In general, there is an increased activity of the complement system during pregnancy, which is essential for maintaining the host’s defense and fetal survival. Unbalanced or excessive activation of the complement system in the placenta is associated with pregnancy complications, such as miscarriage, preeclampsia, and premature birth. Nonetheless, the actual clinical value of monitoring the activation of the complement system during pregnancy remains to be investigated. Unfortunately, normal reference values specifically for pregnant women are missing, and for umbilical cord blood (UCB), data on complement protein levels are scarce. Herein, complement protein analyses (C1q, C3, C4, C3d levels, and C3d/C3 ratio) were performed in plasma samples from 100 healthy, non-medicated and non-smoking pregnant women, collected during different trimesters and at the time of delivery. In addition, UCB was collected at all deliveries. Maternal plasma C1q and C3d/C3 ratio showed the highest mean values during the first trimester, whereas C3, C4, and C3d had rising values until delivery. We observed low levels of C1q and C4 as well as increased C3d and C3d/C3 ratio, particularly during the first trimester, as a sign of complement activation in some women. However, the reference limits of complement analyses applied for the general population appeared appropriate for the majority of the samples. As expected, the mean complement concentrations in UCB were much lower than in maternal plasma, due to the immature complement system in neonates.

## 1. Introduction

The highly conserved complement system has pivotal roles in innate immunity, e.g., by mediating opsonization, lysis, inflammation, and elimination of potential pathogens. Furthermore, it provides a link between the innate and the adaptive immune system by clearance of immune complexes and apoptotic cells [1]. In addition, the complement system is important in normal pregnancy and normal placentation for maternal host defense and fetal survival. Several recent studies suggest that a delicate regulation of complement activation is crucial for successful pregnancy [2,3]. During normal pregnancy, the mother’s immune system undergoes certain adaptions, e.g., downregulation of polymorphic classical class I human leukocyte antigen (HLA) molecules (HLA-A and HLA-B) in invasive trophoblasts [4,5], a shift from the T helper (Th) 1 to the Th2 phenotype [6], and expression of uterine natural killer cells [7] at the maternal–fetal interface to protect the semiallogeneic fetus and placenta [8].

The complement system is organized into three activation pathways: the classical pathway (CP), the lectin pathway (LP), and the alternative pathway (AP). All pathways have different recognition molecules and lead to increased formation of C3 and C5 convertases, resulting in the cleavage of C3 and C5, respectively, and the release of the anaphylatoxins C3a and C5a. Activation of C5 can further elicit assembly of the terminal pathway of the complement cascade to generate the membrane attack complex. C1q is the first molecule of the classical pathway of complement activation and is widely distributed in the human decidual stroma [9,10]. Adequate C1q expression at the maternal–fetal interface appears to be important in placental formation and pregnancy maintenance [9,11]. In both CP and LP, the classical/lectin C3 convertase is produced via cleavage of C4, and the plasma levels of consumed C4 reflect the potential activation of these two pathways. Apoptosis and release of free deoxyribonucleic acid (DNA) occur during placental formation in early pregnancy, which can trigger complement activation and complement system-mediated clearance [2,3]. Three complement regulatory proteins expressed on syncytiotrophoblasts can ensure the process of physiological placentation: decay-accelerating factor, membrane co-factor protein, and protectin (CD59) [12,13]. 

In general, moderate activation of the complement system is crucial for maintaining pregnancy, and any deviation from normal activation and regulation of the complement system can result in adverse pregnancy outcomes (APO), such as recurrent miscarriage [14], premature birth [15], and preeclampsia [16,17,18]. 

The complement components are synthesized from an early stage in the fetus with a relative deficiency of most of the complement proteins in umbilical cord blood (UCB) compared to levels in adults [19,20,21,22]. The first components to be detected are C3 and C4 at weeks 5 and 8, respectively, of gestation, while factor B is present in the fetal circulation at approximately the 10th gestational week (GW) [23,24]. Previous studies assume that all the complement components can be detected at 18–20 weeks of gestation [23,25]. However, more in-depth and recent investigations on the intrauterine development and function of the complement system are scarce [22]. 

To our knowledge, normal ranges for complement proteins during pregnancy are not known. Such intervals would facilitate an increased understanding of the complement system during pregnancy, as well as the clinical assessment of pregnant women, to estimate the risk of APO, e.g., in women with systemic lupus erythematosus (SLE) where the risks of APO are increased [26]. Similarly, there is a knowledge gap regarding the complement activation in UCB [22,27]. 

Most previous studies on the dysregulation of the complement system and APO had a cross-sectional design. The present investigation is a longitudinal study that aims to establish the normal plasma concentration ranges of complement components in healthy women during different stages of normal pregnancy and in the UCB of healthy newborns. In addition, we aimed to investigate potential correlations between the levels of the complement components and some easily assessed variables, such as the mother’s age, parity, body mass index (BMI), levels of serum albumin and cystatin C (cys C), as well as newborns birth weight.

## 2. Methods

### 2.1. Study Population

One hundred healthy, non-medicated and non-smoking women with 100 normal singleton pregnancies were recruited during the years 2015–2018, as previously described [28]. EDTA plasma samples were made available from the pregnancy biobank (Graviditetsbiobanken, GraBB), founded at the Linköping University Hospital in collaboration with Linköping University. Longitudinal clinical data and plasma samples were collected for all the recruited females prospectively during the first (GW 8–11) and second (GW 12–25) trimesters (T1 and T2), as well as at time of delivery (partus). Furthermore, all newborns’ UCB were available. 

The women were aged 18–40 at the time of conception and their baseline BMI was in the range of 18–25 kg/m^2^. All the pregnancies were term (duration ranging from 37–41 weeks) and without complications. Above 90% of the women were of Caucasian origin. Only pregnancies resulting in non-instrumental vaginal delivery and neonates with normal weight for gestational age were considered for the study. In addition, pregnancies were selected by date of birth and evenly distributed over quarters of the year to minimize the impact of seasonal variations. Selected characteristics of the participating women are detailed in Table 1.

### 2.2. Routine Laboratory Analyses

To rule out impaired renal function and/or hypoalbuminemia, cys C and albumin were determined in all plasma samples from all subjects during pregnancy. Albumin and cys C were analyzed at the unit for Clinical Chemistry at Linköping University Hospital. The estimated glomerular filtration rate (eGFR) based on cys C was also calculated [29]. These laboratory variables are reported in Table 1.

### 2.3. Complement Analyses

The complement protein analysis was performed at the unit for Clinical Immunology at Uppsala University Hospital. C1q was quantified by magnetic bead-based immunoassay, which is an in-house assay performed using the Amine Coupling Kit (BIO-RAD) according to the manufacturer’s instructions [30]. The concentrations of C3 and C4 were quantified by nephelometry using an IMMAGE nephelometer (Beckman Coulter, Bromma, Sweden). C3d was measured by nephelometry, after the removal of high molecular weight forms of C3 by polyethylene glycol precipitation [31]. The normal reference limits applied in clinical routine are: C1q = 70–300 mg/L; C3 = 0.67–1.29 g/L; C4 = 0.13–0.32 g/L; C3d < 5.3 g/L; C3d/C3 ratio < 5.3.

### 2.4. Statistics

Statistical analyses were performed with SPSS statistics version 28 (IBM, Armonk, NY, USA) or with GraphPad Prism version 9 (GraphPad Software, La Jolla, CA, USA). As data were mainly normally distributed, the mean values of complement proteins were calculated, and parametric statistical methods were applied. One-way ANOVA was used to detect any significant differences between the groups. To assess the six different comparisons between the different blood samplings (T1, T2, partus, and UCB), paired *t*-tests were performed. The multiple comparisons between the time samples were Bonferroni-adjusted. *p*-values ≤ 0.05 were considered statistically significant. Correlation analyses between continuous variables were performed with Pearson’s r test for parametric data, and the associations of continuous variables and categorical variables were analyzed by the Mann–Whitney *U* test.

### 2.5. Ethical Approval

Oral and written informed consent was obtained from all study participants during their first visit to the antenatal clinic, which allows biobank access to plasma samples as described above, as well as present and future medical records for research purposes. This information was given according to the Declaration of Helsinki. The study protocol was approved by the Swedish Ethical Review Authority (Decision number 2010/296–31 and 2019–00424).

## 3. Results

### 3.1. Complement Protein Levels during Pregnancy 

The current normal complement protein ranges used in clinical routine were compared with the levels detected during pregnancy. Our results show that the mean values of C1q, C3, C4, C3d, and C3d/C3 ratio are within the established normal limits, even with respect to UCB (Table 2). However, during maturity of the pregnancies, complement factors changed. For example, the 5th percentile of C1q was clearly lower than the lower reference limit at all three samplings (T1, T2, and partus). The same observation was made for C4 in T1. In addition, the 95th percentile of both C3d and C3d/C3 ratio showed higher levels than the cut-offs applied in clinical routine for all sampling occasions. All complement proteins levels in UCB plasma were significantly lower than in maternal plasma whereas the C3d/C3 ratio was higher (Table 2 and Figure 1A–E).

### 3.2. Associations between Complement Proteins and Evaluated Variables

C1q at T1 showed significant association with parity, as the detected levels were significantly lower in primipara than in multipara (mean C1q primipara 45.1 vs. 59.0 mg/L; *p* = 0.02). Albumin inversely correlated with C3 and C4 during every studied trimester, and with C3d/C3 ratio in T1. Throughout the pregnancies, albumin levels showed inverse correlations with C3 (*p* < 0.0001), C4 (*p* < 0.0001), C3d/C3 ratio (*p* < 0.0001), and C1q (*p* = 0.02), as demonstrated in Figure 2A–E. As shown in Table 3, BMI measured during the three pregnancy periods was significantly correlated with C3 (*p* = 0.04) and C4 (*p* = 0.03) at T2, and with C3d at T1 (*p* = 0.05). The birth weight of the newborns showed a significant correlation with C3 in UCB (*p* = 0.03) and with C3d at T2 (*p* = 0.05). 

No significant correlations were found between complement levels and cys C and duration of pregnancy (weeks). Neither did we observe any association between complement and the newborn’s sex (Table 3).

## 4. Discussion

The complement proteins investigated herein are quantified in daily clinical routine, e.g., to detect complement deficiencies and complement activation after the formation of immune complexes, as seen in systemic inflammatory diseases, such as SLE and antiphospholipid syndrome [32,33,34]. The regulation of complement activation is a delicate balance between activation and inhibition that is necessary for a normal pregnancy, while unregulated complement activation has a role in the development of pregnancy complications such as miscarriage, preeclampsia, and preterm births [14,15,16].

In this study, we observed that the levels of complement proteins changed significantly during the different stages of pregnancy, but still mainly remained within the normal reference range used in clinical routine. However, as a sign of activation of the CP, decreased levels of C1q (all trimesters), C4 (T1), and increased C3d and C3d/C3 ratio (all trimesters) were observed in approximately 5% of the women. Variation of circulating complement protein levels during the trimesters of normal successful pregnancies probably reflects the delicate regulation of the complement system. A recent study made similar observations and concluded that the pregnancy itself may affect complement protein levels [35].

There is a lack of knowledge regarding the levels of complement proteins in UCB and only a few prior studies have investigated this matter in detail [19,27]. Consistent with the results from Johnson et al. [27], we found significantly lower levels of complement proteins in UCB than in the corresponding maternal plasma, due to the low production of these proteins related to immaturity of the complement system in neonates [22], which poses a considerable risk for severe infections [7,36,37].

Plasma albumin correlated repeatedly and significantly in all periods of pregnancy with both C3 and C4 levels and with C3d/C3 ratio in T1. In addition, albumin levels were slightly below the normal limit during T1 and T2. It is well established that plasma albumin is a known negative acute phase reactant that decreases during infectious and inflammatory conditions [38]. The pregnancy itself normally causes low-grade inflammation, generally represented by elevated levels of C-reactive protein (CRP) in normal and uncomplicated pregnancies [39,40,41]. A recent investigation, performed by our research group on the same study population, found an inverse correlation between plasma albumin and CRP, and CRP levels seemed to be above the normal range limits used for non-pregnant women [28]. In this context, it is worth mentioning that CRP has the potential to activate the CP [42]. However, the positive correlations between plasma albumin and some levels of the complement proteins support a possible association between low-grade inflammation and complement activation during normal pregnancies. 

Cys C in plasma was analyzed herein since it can be used for close supervision and early diagnosis of renal impairment in pregnant women—with better reliability than plasma creatinine according to previous studies [43,44]. Cys C levels in our study population were within the normal limits during T1 and T2 with a slight increase in partus samplings as shown in Table 1. Consequently, the eGFR calculated from cys C showed values within reference limits in T1 and T2, while the mean value of eGFR in partus samplings was below normal limits (eGFR >90 mL/min/1.73 m^2^). As previously mentioned, cys C levels were not found to be decreased in term pregnancy, though GFR of low molecular mass substances is known to increase by at least 40% during pregnancy in general [44]. This is in line with a known decrease in GFR even in normal and healthy term pregnancy, which is associated with glomerular endothelial swelling (endotheliosis) [45,46]. Thus, despite cys C changes during studied pregnancies here, it had not correlated with the complement proteins levels in all studied pregnancy periods. 

The mothers’ BMI was within the normal limits (18–25 kg/m^2^) at the time of conception. BMI was measured at different stages of pregnancy, and it correlated with C3, C4, and C3d during some of the pregnancy periods (Table 3). A positive correlation was also observed between the weight of the offspring and C3 in UCB as well as with C3d in maternal plasma during T2. The relation between body weight or weight changes and levels of AP components of the complement system has previously been highlighted, possibly due to the production of pro-inflammatory factors in adipose tissue or at other sites [47].

The main limitation of this study is the relatively small number of included subjects. The homogeneity of the study population, which was mainly of Caucasian origin, also limits the possibility of generalization of our results to other ethnicities. The longitudinal design of the study, with measurement of complement factors in different stages of pregnancy in a well-characterized study population, represents a major strength of our investigation.

## 5. Conclusions

With the exceptions described above, we conclude that the reference ranges of complement factors used in clinical practice are principally reliable and broad enough to encompass the variations seen in women during normal pregnancy and in newborns after normal gestation and uncomplicated deliveries. Nonetheless, the significant differences observed in levels of complement proteins over time may suggest the implementation of more stringent ranges during different phases of gestation. Our findings have implications, particularly in clinical settings where monitoring of the complement system may be crucial to avoid APO. As expected, complement levels were generally lower in UCB than in maternal plasma due to the still immature neonatal immune system.

## Figures and Tables

**Figure 1 jcm-11-03611-f001:**
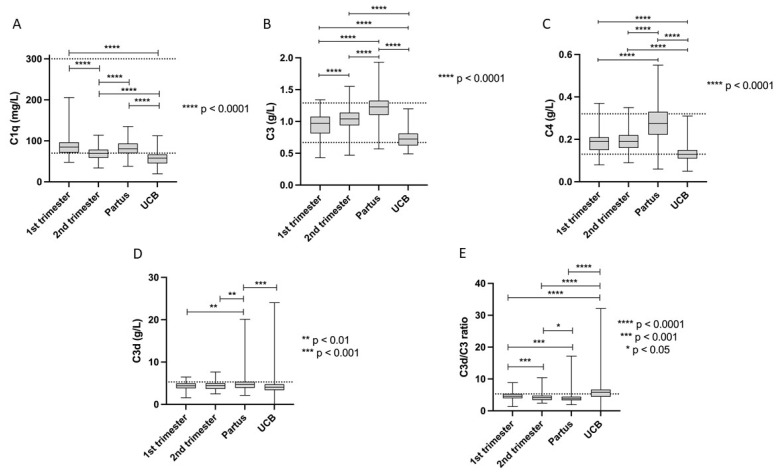
(**A**–**E**) Complement protein levels in maternal plasma during different stages of pregnancy and in umbilical cord blood (UCB). Dotted lines indicate reference limits used in clinical routine. Boxes show the 25th to 75th percentile, with median values marked inside. The provided *p* values are Bonferroni-adjusted.

**Figure 2 jcm-11-03611-f002:**
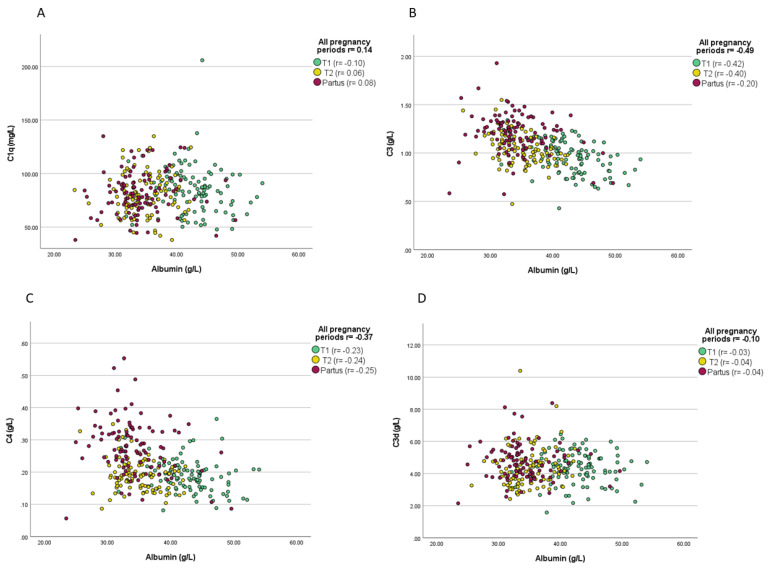
(**A**–**E**) Correlations of plasma albumin levels and different complement levels throughout the pregnancies. Correlation coefficients and significance levels are given both for merged data and for all sampling occasions.

**Table 1 jcm-11-03611-t001:** Characteristics of the participating women ^1^ (*n* = 100) and their newborns.

Mean age, years (SD)	30.0 (3.8)
Caucasian ethnicity, *n* (%)	92 (92)
Mean body mass index at inclusion, kg/m^2^ (SD)	22.0 (1.8)
Mean gestation duration in weeks + days (range)	40 + 1 (37 + 4 − 41 + 4)
Multiparous, *n* (%)	39 (39)
Births in January–March, *n* (%)	25 (25)
Births in April–June, *n* (%)	23 (23)
Births in July–September, *n* (%)	27 (27)
Births in October–December, *n* (%)	25 (25)
Mean birth weight, g (SD)	3572.2 (399.5)
Female newborns, *n* (%)	50 (50)
***Routine Laboratory Analyses*** (mean (SD))	
**T1:** P-albumin (g/L)	42.9 (4.3)
**T2:** P-albumin (g/L)	34.7 (3.6)
**Partus:** P-albumin (g/L)	34.1 (4.5)
**T1:** P-cystatin C (mg/L)	0.74 (0.10)
**T2:** P-cystatin C (mg/L)	0.84 (0.11)
**Partus:** P-cystatin C (mg/L)	1.3 (0.28)
**T1:** eGFR (mL/min/1.73 m^2^)	116.8 (19.2)
**T2:** eGFR (mL/min/1.73 m^2^)	101.3 (17.3)
**Partus:** eGFR (mL/min/1.73 m^2^)	60.1 (13.8)

eGFR, estimated glomerular filtrations rate; P-, plasma; SD, standard deviation; T1, trimester 1; T2, trimester 2. ^1^ The present study population was selected from a larger pregnancy cohort (GRABB) in order to include only healthy females and normal singleton pregnancies [28].

**Table 2 jcm-11-03611-t002:** Complement protein levels in maternal plasma and umbilical cord blood.

Complement	Sampling Occassion	Mean	SD	Min	Max	Percentiles					
						*1st*	*3rd*	*5th*	*95th*	*97th*	*99th*
**C1q** (mg/L) Ref. 70–300	*T1*	84.6	22.3	47.8	205.8	47.8	50.3	52.1	112.2	122.8	205.1
	*T2*	69.8	15.5	34.1	113.9	34.1	39.8	45.8	95.7	103.9	113.9
	*Partus*	81.5	19.7	38.0	134.9	38.0	44.5	46.9	121.8	123.9	134.8
	*UCB*	58.2	16.4	19.9	112.4	19.9	34.2	36.7	85.4	102.3	112.4
**C3** (g/L) Ref. 0.67–1.29	*T1*	0.95	0.17	0.43	1.3	0.43	0.65	0.67	1.2	1.2	1.3
	*T2*	1.1	0.16	0.47	1.6	0.47	0.81	0.82	1.3	1.4	1.6
	*Partus*	1.2	0.21	0.57	1.9	0.57	0.69	0.79	1.5	1.6	1.9
	*UCB*	0.74	0.15	0.49	1.2	0.49	0.52	0.54	1.0	1.1	1.2
**C4** (g/L) Ref. 0.13–0.32	*T1*	0.19	0.05	0.08	0.4	0.08	0.10	0.11	0.29	0.30	0.36
	*T2*	0.19	0.05	0.09	0.4	0.09	0.11	0.13	0.30	0.33	0.35
	*Partus*	0.28	0.09	0.06	0.6	0.06	0.11	0.13	0.41	0.49	0.55
	*UCB*	0.13	0.04	0.05	0.3	0.05	0.08	0.09	0.20	0.21	0.31
**C3d** (g/L) Ref. < 5.3	*T1*	4.4	0.98	1.6	6.5	1.6	2.3	2.6	6.0	6.1	6.5
	*T2*	4.4	1.0	2.5	7.7	2.5	2.8	2.9	6.1	6.4	7.7
	*Partus*	4.9	1.9	2.2	20.1	2.2	2.8	2.9	7.6	8.1	20.1
	*UCB*	4.4	2.6	0.01	24.0	0.02	1.2	1.7	8.2	8.4	23.9
**C3d/C3** ratio Ref. < 5.3	*T1*	4.7	1.2	1.4	8.9	1.4	2.4	3.0	6.7	7.4	8.9
	*T2*	4.3	1.2	2.4	10.4	2.4	2.8	2.9	6.2	6.7	10.4
	*Partus*	4.1	1.7	1.9	17.2	1.9	2.6	2.7	6.6	7.8	17.2
	*UCB*	6.0	3.5	0.01	32.1	0.03	1.8	2.6	9.9	14.0	32.0

C3, complement protein 3; C4, complement protein 4; Ref., reference limits; SD, standard deviation; T1, trimester 1; T2, trimester 2; UCB, umbilical cord blood.

**Table 3 jcm-11-03611-t003:** Associations between complement protein levels and studied variables.

Comp-Lement	Sampling Occasion	Body Mass Index (Mothers’)			P-Cystatin C			Mothers’ Age	Newborns’ Weight	Parity
		*T1*	*T2*	*Partus*	*T1*	*T2*	*Partus*			
**C1q** (mg/L)	*T1*	n.s.	n.t.	n.t.	n.s.	n.t.	n.t.	n.s.	n.s.	***p* = 0.02**
	*T2*	n.t.	n.s.	n.t.	n.t.	n.s.	n.t.	n.s.	n.s.	n.s.
	*Partus*	n.t.	n.t.	n.s.	n.t.	n.t.	n.s.	n.s.	n.s.	n.s.
	*UCB*	n.t.	n.t.	n.t.	n.t.	n.t.	n.t.	n.t.	n.s.	n.s.
**C3** (g/L)	*T1*	n.s.	n.t.	n.t.	n.s.	n.t.	n.t.	n.s.	n.s.	n.s.
	*T2*	n.t.	**r = 0.21**; ***p* = 0.04**	n.t.	n.t.	n.s.	n.t.	n.s.	n.s.	n.s.
	*Partus*	n.t.	n.t.	n.s.	n.t.	n.t.	n.s.	n.s.	n.s.	n.s.
	*UCB*	n.t.	n.t.	n.t.	n.t.	n.t.	n.t.	n.t.	**r = 0.22**; ***p* = 0.03**	n.s.
**C4** (g/L)	*T1*	n.s.	n.t.	n.t.	n.s.	n.t.	n.t.	n.s.	n.s.	n.s.
	*T2*	n.t.	**r = 0.23**; ***p* = 0.03**	n.t.	n.t.	n.s.	n.t.	**r = −0.24**; ***p* = 0.02**	n.s.	n.s.
	*Partus*	n.t.	n.t.	n.s.	n.t.	n.t.	n.s.	n.s.	n.s.	n.s.
	*UCB*	n.t.	n.t.	n.t.	n.t.	n.t.	n.t.	n.t.	n.s.	n.s.
**C3d** (g/L)	*T1*	**r = 0.24**; ***p* = 0.02**	n.t.	n.t.	n.s.	n.t.	n.t.	n.s.	n.s.	n.s.
	*T2*	n.t.	n.s.	n.t.	n.t.	n.s.	n.t.	n.s.	**r = 0.20**; ***p* = 0.05**	n.s.
	*Partus*	n.t.	n.t.	n.s.	n.t.	n.t.	n.s.	n.s.	n.s.	n.s.
	*UCB*	n.t.	n.t.	n.t.	n.t.	n.t.	n.t.	n.t.	n.s.	n.s.
**C3d/C3** ratio	*T1*	n.s.	n.t.	n.t.	n.s.	n.t.	n.t.	n.s.	n.s.	n.s.
	*T2*	n.t.	n.s.	n.t.	n.t.	n.s.	n.t.	n.s.	n.s.	n.s.
	*Partus*	n.t.	n.t.	n.s.	n.t.	n.t.	n.s.	n.s.	n.s.	n.s.
	*UCB*	n.t.	n.t.	n.t.	n.t.	n.t.	n.t.	n.t.	n.s.	n.s.

BMI, body mass index; C3, complement protein 3; C4, complement protein 4; n.s., not significant; n.t., not tested; T1, trimester 1; T2, trimester 2; UCB, umbilical cord blood

## Data Availability

Data available on request from the authors.

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
