# Peer review of "Variation of Complement Protein Levels in Maternal Plasma and Umbilical Cord Blood during Normal Pregnancy: An Observational Study"

_jcm, 2022, doi:10.3390/jcm11133611_

Round 1

Reviewer 1 Report

Dear Authors, thank you very much for this interesting and important piece o work. I have read it with a great interest. The manuscript is well written. Nevertheless, some minor amendments are required. 

The title and the abstract clearly presents the study.

The introduction part gives an appropriate background information and explains the study rationale.

The methods' section is detailed. However, there are some structure inaccuracy found: The data given in lines 103-110 as well as Table 1, must be moved to the Results section. 

The results' section should begin with the study subjects' description, i.e. the info from lines 103-110 and table 1 should be presented here. The overall results section is well described and detailed with the sufficient figures and tables. 

The study results possible clinical implication should be mentioned in the discussion section.

Author Response

We are grateful for the careful review of our manuscript.

Regarding the first amendent; in fact, our pregnancy cohort was not collected by us. Instead, the females were selected from a larger Pregnancy cohort (GRABB) in order include only healthy mothers and "normal pregnancies" (Wirestam L, et al. Front Immunol 2021;12:722118). For instance, we sought to included births evenly during the year as well as an even sex ratio among newborns. Based on this, we consider that the selection of women was not a part of the Results neither were the background variables. Thus, we prefer to keep this section and Table 1 in Materials & Methods.

Regarding the second amendent, we agree. We have highlighted the clinical implication in the Conclusions.

Reviewer 2 Report

Muna Saleh et al. provide an intersting observational study about complement protein levels in maternal Plasma and umbilical cord Blood during normal pregnancy.

The manuscript is well written and organized. However, I have some minor concerns:

1) The table 3 is difficult to read and is partially redundant as figure 2 shows some correlations too. It would be better to insert the significative correlations in the dedicate section of the text, indicating r and p values.

2) Please improve the resolution of figure 2.

Author Response

We are grateful for the careful review of our manuscript.

According to the Reviewer's suggestion, we revised Table 3 and excluded Plasma Albumin as these data are shown in Figure 2.

The resolution of Figure 2 has been substantially improved.